# Dexamethasone-Induced Mitochondrial Dysfunction and Insulin Resistance-Study in 3T3-L1 Adipocytes and Mitochondria Isolated from Mouse Liver

**DOI:** 10.3390/molecules24101982

**Published:** 2019-05-23

**Authors:** Guangxiang Luan, Gang Li, Xiao Ma, Youcai Jin, Na Hu, Ji Li, Zhenhua Wang, Honglun Wang

**Affiliations:** 1Key Laboratory of Tibetan Medicine Research, Northwest Institute of Plateau Biology, Chinese Academy of Sciences, Xining 810008, China; gxluan@nwipb.cas.cn (G.L.); xma@nwipb.cas.cn (X.M.); jinyc04@mails.ucas.ac.cn (Y.J.); huna@nwipb.cas.cn (N.H.); 2State Key Laboratory of Plateau Ecology and Agriculture, Qinghai University, Xining 810008, China; ligang@ytu.edu.cn; 3Center for Mitochondria and Healthy Aging, College of Life Sciences, Yantai University, Yantai 264005, China; lijimelissafang@yahoo.com; 4Key Laboratory of Tibetan Medicine Research of Qinghai Province, Northwest Institute of Plateau Biology, Chinese Academy of Sciences, Xining 810008, China; 5University of Chinese Academy of Sciences, Beijing 100049, China

**Keywords:** dexamethasone, insulin resistance, 3T3-L1 adipocytes, mitochondrial dysfunction, isolated mitochondria

## Abstract

Dexamethasone is a glucocorticoid analog, which is reported to induce insulin resistance and to exacerbate diabetic symptoms. In this study, we investigated the association between mitochondrial dysfunction and the pathophysiology of dexamethasone-induced insulin resistance. An insulin resistance model in 3T3-L1 adipocyte was established by 48-h treatment of 1 μM dexamethasone, followed with the detection of mitochondrial function. Results showed that dexamethasone impaired insulin-induced glucose uptake and caused mitochondrial dysfunction. Abnormality in mitochondrial function was supported by decreased intracellular ATP and mitochondrial membrane potential (MMP), increased intracellular and mitochondrial reactive oxygen species (ROS) and mtDNA damage. Mitochondrial dynamic changes and biogenesis were suggested by decreased *Drp1*, increased *Mfn2*, and decreased *PGC-1*, *NRF1*, and *TFam*, respectively. The mitochondrial DNA (mtDNA) copy number exhibited no change while the mitochondrial mass increased. In agreement, studies in isolated mitochondria from mouse liver also showed dexamethasone-induced reduction of mitochondrial respiratory function, as suggested by decreased mitochondrial respiration controlling rate (RCR), lower MMP, declined ATP synthesis, opening of the mitochondrial permeability transition pore (mPTP), damage of mtDNA, and the accumulation of ROS. In summary, our study suggests that mitochondrial dysfunction occurs along with dexamethasone-induced insulin resistance in 3T3 L1 adipocytes and might be a potential mechanism of dexamethasone-induced insulin resistance.

## 1. Introduction

Insulin resistance, characterized by a dampened response to insulin signals and decreased uptake and/or utilization of glucose in the target tissue, is a major cause of type 2 diabetes [1,2]. Dexamethasone is a glucocorticoid analog, and often exacerbates diabetic symptoms [3,4], which has been supported by both clinical studies and studies in animals or cultured cells [4,5]. Dexamethasone may inhibit the translocation of GLUT4 to the cell membrane [6,7,8] to blunt the insulin-induced glucose uptake in the target tissue [9]. 

A large number of studies have shown that the occurrence of diabetes is closely related to mitochondrial dysfunction [10,11]. The expression of markers of mitochondrial capacity was decreased in fat tissue of ob/ob mice [12]. Compared with normal people, patients with type 2 diabetes contain fewer mitochondria in muscles [13]. Nuclear magnetic resonance spectroscopy studies have shown that mitochondrial oxidative phosphorylation function in muscle and liver tissues of individuals with insulin resistance was significantly reduced [14]. Mitochondrial DNA undergoes a high rate of mutation because of its bare DNA, imperfect DNA repair machinery, and extensive exposure to reactive oxygen species (ROS) [15,16]. ROS in cells are mainly produced by the electron transport complex on the electron transport chain (ETC) and a high level of ROS can directly damage mitochondria [17]. Studies suggest the role of mitochondria-generated ROS in the pathophysiology of diabetes [11]. All this evidence points to the contribution of mitochondrial dysfunction in insulin resistance. 

Adipose tissue is one of the major targets for insulin and adipose tissue dysfunction usually leads to insulin resistance. A prospective cohort study in women found that in more than 60% of the diabetes patients their condition was related to obesity [18]. Phosphorylation and deactivation of the insulin receptor caused by a high level of fatty acids may underlie the suppressed insulin signaling and reduced glucose uptake in target tissues under obesity conditions [19]. Based on the information above, this study aimed to determine whether dexamethasone treatment affects the mitochondrial function especially in adipocytes, which might help our understanding of the mechanism of dexamethasone inducing insulin resistance and benefit its clinical use. 

## 2. Results

### 2.1. Effects of Dexamethasone on Insulin-Induced Glucose Uptake and AKT Phosphorylation Level in 3T3-L1 Adipocytes

Insulin resistance could be induced by dexamethasone in cultured 3T3-L1 adipocytes, which is widely used as an in vitro model the study of insulin resistance and type 2 diabetes mellitus [9]. However, the duration of dexamethasone treatment is inconsistent in the literature [4,20]. In this section, we first determined the effect of treatment time of dexamethasone by detecting insulin-induced 2-(*N*-(7-Nitrobenz-2-oxa-1,3-diazol-4-yl)Amino)-2-Deoxyglucose (2-NBDG) uptake and protein kinase B (PKB/AKT) phosphorylation. As shown in Figure 1, 48 and 72 h of dexamethasone (1 µM) exposure caused a significant decrease in insulin-induced 2-NBDG uptake and AKT phosphorylation in adipocytes, compared to the untreated group. In contrast, 24 h incubation did not induce changes in 2-NBDG uptake or ATP phosphorylation. Thus, the treatment time of 48 h with dexamethasone was selected to establish the insulin resistance model in the following studies. 

### 2.2. Dexamethasone Treatment-Induced ROS Accumulation

A large amount of evidence points out that the occurrence of type 2 diabetes is closely related to oxidative stress. Due to the major roles of mitochondria in both generation and elimination of cell oxidative stress. Therefore, in this study, we monitored the production of intracellular and mitochondrial ROS in dexamethasone-treated and control adipocytes using 2′,7′-Dichlorodihydrofluorescein diacetate (DCFH-DA) and Mito-SOX probes, respectively. As shown in Figure 2a,b, ROS levels in dexamethasone-treated 3T3-L1 adipocytes was dramatically increased. Similar to intracellular ROS, mitochondrial ROS of dexamethasone-treated adipocytes was also significantly elevated (Figure 2c).

### 2.3. Dexamethasone Reduces MMP and ATP Production

Cellular ATP and mitochondrial membrane potential (MMP) were important parameters in reflecting the function of mitochondria. Treatment with dexamethasone caused a marked decrease in both adenosine triphosphate (ATP) (Figure 3a) and MMP in 3T3-L1 (Figure 3b,c) adipocytes. 

### 2.4. Effects of Dexamethasone on Mitochondrial Mass and mtDNA

Mitochondrial mass and mtDNA, which might be altered by ROS, was also detected. As shown in Figure 4a,b, dexamethasone treatment caused an increase of mitochondrial mass, while the mtDNA copy number was not altered (Figure 4b). Results from a long PCR experiment (Figure 4c) showed reduced polymerase chain reaction (PCR) products of long fragments in a dexamethasone-treated group, while short fragments showed no difference between the two groups, which indicated that mtDNA was damaged by dexamethasone.

### 2.5. Dexamethasone Treatment Affects Mitochondrial Biogenesis and Dynamics in 3T3-L1 Adipocytes

To explore the effect of dexamethasone on mitochondrial biogenesis and dynamics, mRNA levels of *PGC-1α*, *NRF1*, and *TFam*, three critical factors in mitochondrial biogenesis, and *Mfn1*, *Mfn2*, and *Drp1*, three mitochondrial dynamic proteins, were examined with quantitative RT-PCR. The expression level of the Mfn1, Mfn2, and Drp1 was also investigated by western blot. As speculated, dexamethasone treatment caused reduced expression of *PGC-1α* in 3T3-L1 adipocytes. Meanwhile, *NRF1* and *TFam*, as the downstream of *PGC-1α*, were also drastically decreased (Figure 5a). Figure 5b showed that dexamethasone treatment resulted in an obvious increase in *Mfn2* transcription levels (*p* < 0.05), while *Mfn1* expression was not altered. Meanwhile, levels of the *Drp1* were strikingly reduced (*p* < 0.05). Western blot analysis of mitochondrial dynamic factors showed consistent results (Figure 5c).

### 2.6. Effects of Dexamethasone on Mitochondria Isolated from Mouse Liver

Mitochondria were also isolated from mouse liver to further explore the effects of dexamethasone. RCR (ratio of state III and state IV) is an index used to assess respiratory chain function and oxidative phosphorylation. As shown in Figure 6, treatment of mitochondria with dexamethasone decreased RCR dramatically. Moreover, this treatment increased ROS level, dropped MMP, declined ATP synthesis, induced opening of mPTP, and damaged mtDNA (Figure 7). These results indicate that dexamethasone can directly disrupt the structure and function of mitochondria.

## 3. Discussion

Mitochondria are the center of energy supply in cells, providing more than 95% of ATP required for cell metabolism [21]. Studies found mitochondrial dysfunction in a large number of clinical cases of metabolic diseases. Mitochondria are thus a promising target in identifying pathogenesis and treatment strategies for various metabolic diseases.

3T3-L1 pre-adipocytes have the ability to proliferate and differentiate into adipocytes. After being stimulated by specific inducers, they can differentiate into mature adipocytes and thus have been widely used in studies related to glucose and lipid metabolism. Many methods for establishing insulin resistance in adipocytes have been established. To study the association between mitochondrial dysfunction and insulin resistance, we established an insulin resistance model in 3T3-L1 adipocytes using dexamethasone treatment for 48 h, followed by the detection of mitochondrial function. As shown in Figure 2 and Figure 3, insulin resistance was successfully induced by dexamethasone treatment in 3T3-L1 adipocytes. During this process, intracellular levels of ATP and MMP were decreased while the ROS level dramatically increased. Being critical for MMP [22], complexes I and III on the mitochondrial electron transport chain are also major generators of intracellular ROS. This information suggests that, along with the decreased glucose uptake, dexamethasone also causes oxidative stress in mitochondria, decreasing the level of mitochondrial oxidative phosphorylation.

The copy number of mtDNA in each mitochondrion was considered to be constant and thus the total copy number of mtDNA in cells tells the number of mitochondria [23]. The repair mechanism of mtDNA is incomplete and mtDNA is located near the respiratory chain, so mtDNA is more vulnerable when it is exposed to oxidative damage, compared with nuclear DNA. In this study, we investigated the effect of dexamethasone on mtDNA damage and the mtDNA copy number using long PCR and quantitative RT-PCR. Detection of mtDNA damage by long PCR experiments depends on the hampered movement of polymerases and altered replication after DNA damage [10]. After treatment with dexamethasone, long fragment PCR products were significantly reduced, while short fragment products showed no change (Figure 4c), which indicated dexamethasone-induced mtDNA damage. As the mtDNA was damaged by dexamethasone, the synthesis of complex proteins on the mitochondrial electron transport chain will be hindered, resulting in obstructed electron transport. On the other hand, however, the mtDNA copy number was not altered after dexamethasone treatment (Figure 4b). The unaltered mtDNA copy number might be attributed to the elevated in mitochondrial mass (Figure 4a), which might work as a compensatory mechanism to maintain energy supply. 

Mitochondrial biogenesis is a critical adaptation of cells being exposed to chronic energy deprivation [24]. Mitochondrial biogenesis could be regulated by multiple factors such as NRF1 and TFam. NRF1 can promote the transcription of many nuclei-encoded mitochondrial proteins, including those involved in oxidative phosphorylation and respiratory complexes. TFam can enhance DNA replication and gene transcription in mitochondria via directly binding to the mitochondrial genome. As a critical transcriptional co-activator, PGC-1α regulates key factors including NRF1 and TFam and is suggested to promote mitochondrial biogenesis [25]. When the expression of these factors changes, mitochondrial biogenesis will be disordered. As shown in Figure 5a, dexamethasone treatment markedly decreased the transcription levels of *PGC-1α*, *NRF1*, and *TFam* in adipocytes. After dexamethasone treatment, an insufficient energy supply, and decreased mitochondria biogenesis may work synergistically to result in a severe shortage of intracellular energy supply.

It is interesting to note that, although mitochondrial biogenesis is attenuated by dexamethasone, the mitochondrial mass is increased at the same time. A possible explanation for the discrepancy might be the impaired degradation of damaged mitochondria. Under normal conditions, harmful stimuli such as oxidative stress, aging, and energy limitation drive damaged mitochondria to be encapsulated in autophagosomes, fused with lysosomes and finally degraded. Mitochondrial autophagy abnormalities can result in increased accumulation of damaged mitochondria, leading to mitochondrial dysfunction. At the same time, due to severe mtDNA damage, the replication process of mtDNA is blocked, so the number of mitochondrial DNA remained unchanged. While a moderate reduction of *TFam* and *NRF1* might underlie the maintained mtDNA levels, other unknown factors may also contribute to the compensation after dexamethasone treatment and worth further exploration.

In healthy cells, mitochondria undergo dynamic fusion and fission process. This dynamic process is critical for maintaining constant changes in shape, size, and network of mitochondria, which are under the control of regulatory proteins, such as Mfns and DRP1 [26,27]. Our results (Figure 5b,c) showed that the expression of Mfn2 and Drp1 changed in opposite direction after dexamethasone treatment, indicating a disrupted fusion–fission balance of mitochondria. Mfns were suggested to play an important role in mitochondria fusion. We observed the increase of Mfn2 after dexamethasone treatment while no change in Mfn1. Mfn1 was proposed to be more efficient than Mfn2 in fusion of mitochondria and mechanism underlying the differential changes between Mfn1 and Mfn2 worth further investigation.

Studies in isolated mitochondria (Figure 7) showed that dexamethasone caused a large amount of ROS production, which impaired proteins in the electron transport chain and mtDNA, eventually leading to mitochondrial respiratory function damage and more ROS generation. The swelling of isolated mitochondria was due to the opening of the mPTP after dexamethasone treatment in the absence of Ca^2+^. The opening of mPTP may cause flow back of protons from the mitochondrial membrane space to the matrix, thereby reducing the MMP and ATP synthesis and leading to metabolic abnormalities in cells and tissues.

## 4. Materials and Methods

### 4.1. 3T3-L1 Cell Culture and Induction of Insulin Resistance

3T3-L1 cells were purchased from the Shanghai Institute of Biochemistry and Cell Biology (Shanghai, China). In a standard humidified incubator with a set of 5% CO_2_ at 37 °C, cells were cultured in Dulbecco’s Modified Eagle’s Medium (DMEM; Corning, New York, NY, USA) containing 10% fetal bovine serum (FBS; Gibco, Carlsbad, CA, USA) and 4.5 g/L glucose. The procession of 3T3-L1 pre-adipocytes differentiation and insulin resistance induction was performed as previously described [28,29]. Cells were cultured for 2 days before further treatment. Differentiation induction was initiated by induction medium I (DMEM, 10% FBS, 0.5 mM 3-isobuty-1-methylxanthine (IBMX), 1 μM dexamethasone, and 10 μg/mL insulin) for 2 days, and switched to induction medium II (DMEM containing 10% FBS, 10 μg/mL insulin) for 2 days. After induction, cells were then cultured for four days in DMEM containing 10% FBS and 95% of the cells exhibited an adipocyte phenotype. To establish an insulin resistance model, 3T3-L1 adipocytes were treated with 1 µM dexamethasone for 24 h, 48 h, and 72 h, respectively. Insulin resistance of cultured 3T3-L1 adipocytes after dexamethasone treatment was verified by insulin-stimulated glucose uptake and AKT phosphorylation.

### 4.2. Experimental Animals

Five-week-old ICR (Institute of Cancer Research) mice were provided by the Animal Center of Shandong Luye Pharmaceutical Co. Ltd. (Yantai, China) and maintained in a breeding room of 12 h light/dark cycle, with free access to food and water. All experimental protocols involving the usage of animals were approved by the Institutional Animal Care and Use Committee of the Chinese Academy of Sciences, Northwest Institute of Plateau Biology (NWIPB20180309-02).

### 4.3. Preparation of Liver Mitochondria Suspension

As previously described [30], animals were sacrificed by cervical dislocation, and then the liver was quickly removed and placed in an ice-cold isolated buffer (250 mM sucrose, 10 mM Tris-HCl, and 1 mM EDTA, pH 7.4). After trimming, the liver was rinsed and homogenized with a homogenizer in an isolation buffer. In order to preserve the integrity of mitochondria, the entire isolation process was carried out at 4 °C. After centrifugation at 700× *g* for 10 min, the supernatant was collected and centrifuged at 7000× *g* for 10 min again. Then the supernatant was discarded, and the mitochondria pellet was washed by resuspending in 5 mL isolated buffer and centrifuged twice at 7000× *g* for 10 min. A clean mitochondria solution was obtained after suspension in respiration buffer (20 mM sucrose, 100 mM KCl, 10 mM KH_2_PO_4_, 5 mM HEPES, 2 mM MgCl_2_, 1 mM EDTA) to produce a mitochondria suspension with 5 mg/mL protein concentration and placed on ice for immediate use. The BCA assay was used to measure the liver mitochondrial protein concentration.

### 4.4. 2-NBDG Uptake Assay

After 24, 48, and 72 h treatment with dexamethasone, 3T3-L1 adipocytes were incubated with insulin (100 nM) for 15 min followed by incubation of 2-NBDG (100 μM, dissolved in glucose-free culture medium) for another 30min. Cells were washed twice with pre-cold PBS and suspended in 300 μL pre-cold PBS after incubation. To detect fluorescence intensity (485/20 nm for excitation and 540/20 nm for emission) for each measurement, 20,000 single cells were collected using a FACS Aria^TM^ flow cytometer (Becton Dickinson, San Jose, CA, USA) [31].

### 4.5. Western Blot Analysis

Western blot analysis was performed following the manufacturer’s instruction and previous studies [7]. After dexamethasone treatment, 3T3-L1 adipocytes were washed twice with ice-cold PBS and lysed in 100 μL RIPA buffer (Beyotime, Nanjing, China). Homogenates were centrifuged at 12,000 rpm for 15 min at 4 °C and supernatant was collected. Protein concentration was determined by the BCA method. Denatured proteins were loaded to SDS-PAGE gel for separation and transferred to PVDF membrane. The membrane was incubated with a primary antibody overnight at 4 °C after antigen blocking with 5% skim milk for 1 h at room temperature. The membrane was then washed with PBS three times (10 min each) and incubated with horseradish peroxidase-conjugated secondary antibody for 1 h at room temperature. After washing with PBS, the membrane was incubated with chemiluminescent substrate band visualized by a 5200 Multi Luminescent image analyzer (Tanon Science & Technology Co., Ltd. Shanghai, China). The intensity of immunoreactive bands were quantitated by using ImageJ (NIH). The band density of interested proteins was normalized to loading controls.

### 4.6. Determination of ROS Generation

Production of intracellular ROS was determined by 2′,7′-dichlorofluorescein diacetate (DCFH-DA; Sigma, St. Louis, MO, USA), with both confocal laser microscopy and flow cytometry. Cells were cultured on coverslips in 24-well plates and treated with 10 μM DCFH-DA at 37 °C for 30 min in a dark environment. After washing with pre-warmed PBS twice, images were captured with a confocal laser scanning microscope (Olympus, Tokyo, Japan) at 470/530 nm (ex/em). For flow cytometry, cells were seeded in 6-well plates and treated with 10 μM DCFH-DA at 37 °C for 30 min in a dark environment. After washing with pre-warmed PBS twice, cells were suspended in PBS and analyzed with FACS Aria^TM^ flow cytometer (Becton Dickinson, San Jose, CA, USA) at 470/530 nm (ex/em) [32]. To detect the mitochondrial ROS, 5 μM Mito-SOX reagent (Thermo Fisher Scientific, Waltham, MA, USA) was used instead of DCFH-DA using a FACS Aria^TM^ flow cytometer (Becton Dickinson, San Jose, CA, USA) at wavelengths of 510/580 nm (ex/em).

Isolated mitochondrial ROS production was also measured using DCFH-DA. Mitochondria (1 mg/mL of protein concentration) treated with 10 μM dexamethasone were incubated in 0.5 mL of the respiration buffer containing 2.5 mM succinate, 2.5 mM malic acid, 2.5 mM ADP, and 10 µM DCFH-DA for 30 min and then measured by a SpectraMax Paradigm Multi-Mode Microplate Reader (Molecular Devices, CA, USA) at 470/530 nm (ex/em) wavelengths.

### 4.7. ATP Content Measurement

ATP content in 3T3-L1 adipocytes or isolated mitochondria was measured using a luciferase-based luminescence enhanced ATP assay kit (Beyotime, Shanghai, China). Cells were washed twice with ice-cold PBS, harvested in 100 μL ice-cold ATP releasing buffer, and centrifuged at 12,000× *g* for 5 min at 4 °C. Isolated mitochondria (1 mg/mL of protein concentration) were incubated in 0.5 mL of the respiration buffer with 2.5 mM succinate, 2.5 mM malic acid, and 2.5 mM ADP for 10 min. ATP content in cell lysates and the mitochondrial suspension was then determined using a SpectraMax Paradigm Multi-Mode Microplate Reader (Molecular Devices, Sacramento, CA, USA).

### 4.8. MMP Detection

MMP was determined by JC-1 (Sigma, St. Louis, MO, USA). After different treatments, cells were treated with 10 μM JC-1 at 37 °C for 30 min in a dark environment. After being washed twice with pre-warmed PBS and suspended in pre-warmed PBS, the analysis was performed by a FACS Aria^TM^ flow cytometer (Becton Dickinson, San Jose, CA, USA) under an excitation of 488 nm [32]. Isolated mitochondria (1 mg/mL of protein concentration) were incubated in 0.5 mL of the respiration buffer with 2.5 mM succinate, 2.5 mM malic acid, 2.5 mM ADP, and 10 µM JC-1 probe (with or without 10 μM dexamethasone) for 30 min and measured by a SpectraMax Paradigm Multi-Mode Microplate Reader (Molecular Devices, Sacramento, CA, USA) at 488/530, 580 nm (ex/em) wavelengths.

### 4.9. Mitochondria Mass

After treatment, cells were incubated with 10 μM Mito-Tracker Green at 37 °C for 30 min in a dark environment. After that, cells were washed twice with pre-warmed PBS, suspended in pre-warmed PBS, and analyzed with FACS Aria^TM^ flow cytometer (Becton Dickinson, San Jose, CA, USA) at 490/516 nm (ex/em) wavelengths [33].

### 4.10. DNA and RNA Extraction and cDNA Synthesis

Total genome DNA or RNA was isolated with a DNA extraction kit (TIANGEN, Beijing, China) or Trizol reagent (Invitrogen, Carlsbad, CA, USA), respectively. The quality of isolated DNA or RNA was determined by spectrophotometry at 260 nm. Reverse transcription was performed with 1 µg total RNA and M-MLV Reverse Transcriptase Kit (Promega A3500; Promega, Madison, WI, USA). Briefly, 40 μL of total reaction volume was used in a Veriti 96 Well Thermal Cycler long PCR system (Applied Biosystems, Foster City, CA, USA) with the following reaction procedure: 3 min at 72 °C, 90 min at 42 °C, 15 min at 70 °C, and hold at 4 °C.

### 4.11. Real-time Quantitative PCR (RT-PCR)

The transcription level of specific genes and the mtDNA copy number were detected by RT-PCR using templates of cDNA and total genome DNA, respectively. RT-PCR was performed on a Rotor-Gene Q Sequence Detection System (QIAGEN, Germany) using SYBR Premix Ex Taq II (Takara Bio INC) [34]. RT-PCR was performed in 20 μL system (1μL synthetic cDNA + 0.5 μM primers + 10 μL SYBR *Premix Ex Taq* II), with the following procedure: 95 °C, 10 min; 95 °C, 10 s, 40 cycles, 60 °C, 15 s; 72 °C, 20 s; 72 °C, 10 min. The (2^−△△Ct^) value was calculated with β-actin as the inner control [35]. Sequences of primers being used were shown in Table 1.

### 4.12. Long PCR Experiments

Long PCR reactions were performed in Veriti 96 Well Thermal Cycler long PCR system using 2 pairs of primers (Table 1). Long fragment (8636-bp) PCR reactions were performed in a 20 μL system (15 ng templates + 0.5 μM primers + 10 μL Taq 2x Master Mix (NEB, Ipswitch, MA, USA)). The following procedure was used: 94 °C, 2 min, 94 °C, 45 s, 28 cycles, 61 °C, 10 s; 68 °C, 8 min; 68 °C, 7 min. For short fragment (117-bp) PCR reactions, Taq DNA Polymerase (Thermo Scientific™, Waltham, MA, USA) was used and the following procedure was used: 94 °C, 10 min; 94 °C, 10 s, 30 cycles; 65 °C, 15 s; 72 °C, 20 s; 72 °C, 10 min. PCR products were loaded to ethidium bromide-containing agarose gels for separation. The content of each PCR fragment was determined by the intensity of fluorescence under a transilluminator (Tanon Science & Technology Co., Ltd. Shanghai, China), and the ratio of long mtDNA/short mtDNA was calculated [36].

### 4.13. Isolated Mitochondrial Respiratory Test

Isolated mitochondrial respiratory was tested by measurement of oxygen consumption rates using a Clark electrode with 500 μL reaction chamber at 30 °C water bath [37]. The chamber was filled with fresh respiration buffer until a stable baseline is observed. Freshly isolated mitochondria of 500 μg was added to the chamber, followed by a period of waiting until the rate of oxygen consumption stabilized. After succinate (5 mM) was added, a low oxygen consumption rate (state IV) was observed. Then 5 mM ADP was added to stimulate a high oxygen consumption rate (state III), which switched back to the IV rate due to the consumption of added ADP. The respiratory control ratio (RCR) (state III rate/state IV rate) is sensitive and can reflect the ability of mitochondrial oxidative phosphorylation.

### 4.14. Determination of Opening of mPTP

The opening of mPTP was measured by Ca^2+^-induced swelling of isolated liver mitochondria [30]. Isolated mitochondria were re-suspended in swelling buffer (120 mM KCl, 20 mM MOPS, 10 mM Tris–HCl, and 5 mM KH_2_PO_4_, pH 7.4) to a final concentration of 0.5 mg/mL protein. Mitochondria solution was first treated with or without dexamethasone for 3 min at 37 °C. Then 250 μM CaCl_2_ was added to induce mPTP opening for 15 min. After that, absorbance was measured with a SpectraMax Paradigm Multi-Mode Microplate Reader at 540 nm (Molecular Devices, CA, USA).

### 4.15. Statistical Analysis

All data were expressed as the mean ± SD from three independent experiments. Statistical analyses were performed using Student’s *t*-test, one-way of variance (ANOVA), using SPSS 17.0 software.

## 5. Conclusions

In conclusion, our results demonstrated that mitochondrial dysfunction was induced in the process of dexamethasone-induced insulin resistance. During this process, mitochondrial function was impaired, dynamics were altered, and biogenesis was inhibited. ROS overproduction led to mtDNA damage and a significant decrease of MMP followed by a reduction of intracellular ATP. To determine whether dexamethasone induced insulin resistance in 3T3-L1 cells depends on the impaired mitochondrial structure and function, and further studies are warranted.

## Figures and Tables

**Figure 1 molecules-24-01982-f001:**
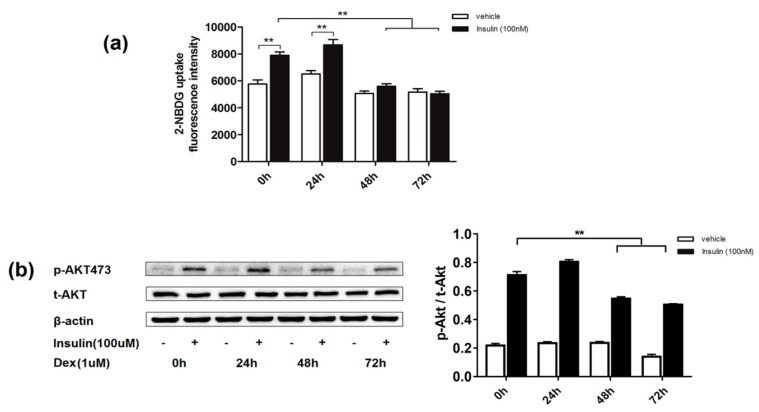
Effect of dexamethasone on 2-NBDG uptake and AKT phosphorylation in 3T3-L1 adipocytes. Glucose uptake was measured by a fluorescent probe of 2-NBDG (**a**). AKT phosphorylation was detected by western blotting. “+”: treatment with insulin; “−“: treatment without insulin (**b**). Data are expressed as Means ± SD, three independent triplicate experiments were performed. ** *p* < 0.01.

**Figure 2 molecules-24-01982-f002:**
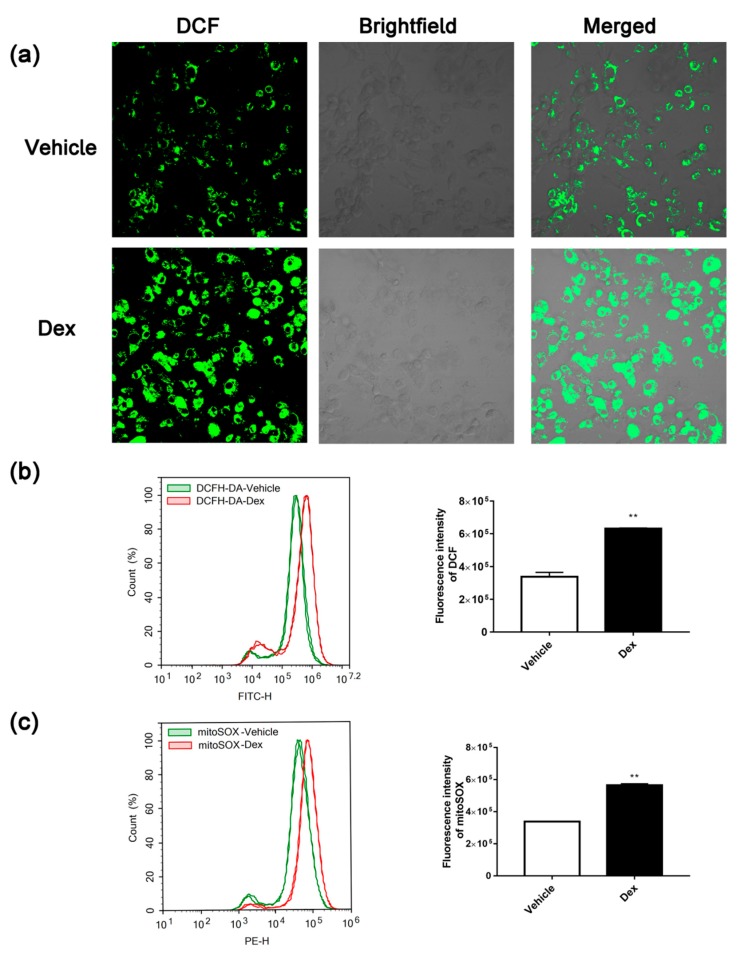
Effect of dexamethasone on intracellular or mitochondrial ROS levels in 3T3-L1 adipocytes. The intracellular ROS were detected by DCFH-DA probe with a confocal laser scanning microscope (200× magnification) (**a**) and a FACS Aria^TM^ flow cytometer (**b**). The mitochondrial ROS were detected by Mito-SOX probe with a FACS Aria^TM^ flow cytometer (**c**). Data are expressed as Means ± SD, three independent triplicate experiments were performed. ** *p* < 0.01.

**Figure 3 molecules-24-01982-f003:**
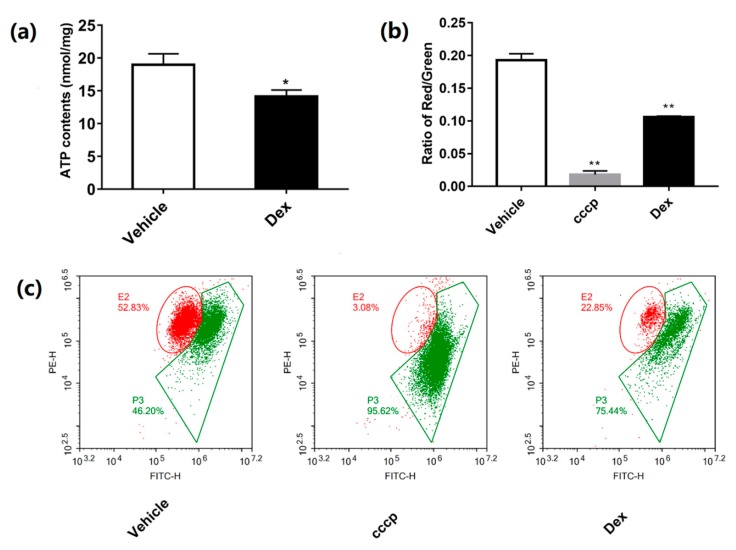
Effect of dexamethasone on mitochondrial dysfunction in 3T3-L1 adipocytes. Level of cellular ATP was determined by luciferase-based luminescence enhanced ATP assay kit (**a**). MMP was measured by FACS Aria^TM^ flow cytometer with a fluorescent probe of JC-1 (**b** and **c**). Data are expressed as Means ± SD, three independent triplicate experiments were performed. * *p* < 0.05 and ** *p* < 0.01.

**Figure 4 molecules-24-01982-f004:**
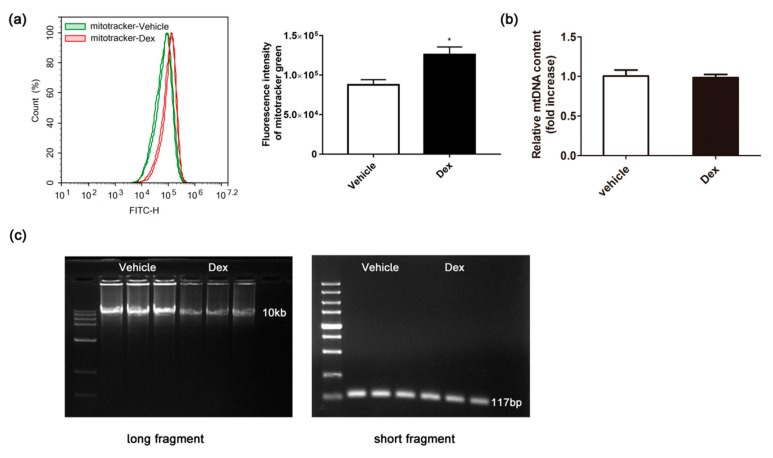
Effect of dexamethasone on mtDNA mass, mtDNA copy number and mtDNA damage in 3T3-L1 adipocytes. The cellular mtDNA mass was detected by FACS Aria^TM^ flow cytometer using a fluorescent probe of the mitotracker green (**a**). The cellular mtDNA content was determined by RT-PCR (**b**). The mtDNA damage was suggested by the ratio of long and short fragments using long PCR (**c**). Data are expressed as Means ± SD, three independent triplicate experiments were performed. * *p* < 0.05.

**Figure 5 molecules-24-01982-f005:**
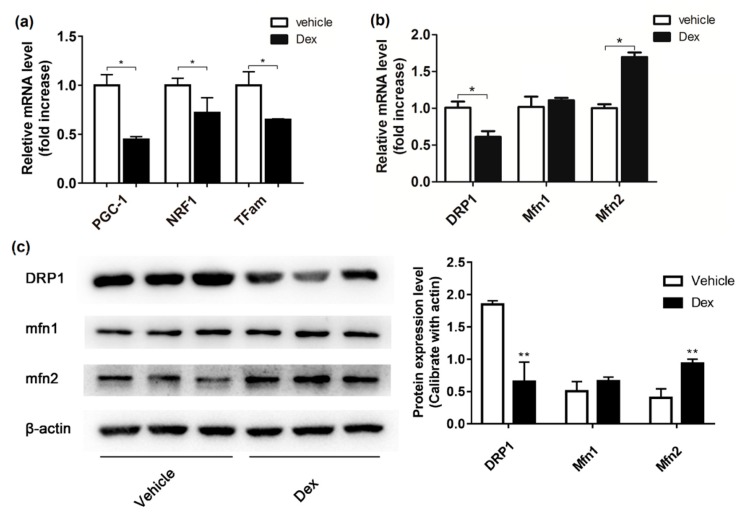
Effect of dexamethasone on mitochondrial dynamics and biogenesis in 3T3-L1 adipocytes. Expression of PGC-1α, *NRF**1*, and *TFam* were determined by reverse transcriptional RT-PCR, with *β-actin* as inner control (**a**). *Mfn1*, *Mfn2*, and *Drp1* transcriptional and expressional level in adipocytes were analyzed by reverse transcriptional RT-PCR (**b**) and western blotting (**c**). Data are expressed as Means ± SD, three independent triplicate experiments were performed. * *p* < 0.05 and ** *p* < 0.01.

**Figure 6 molecules-24-01982-f006:**
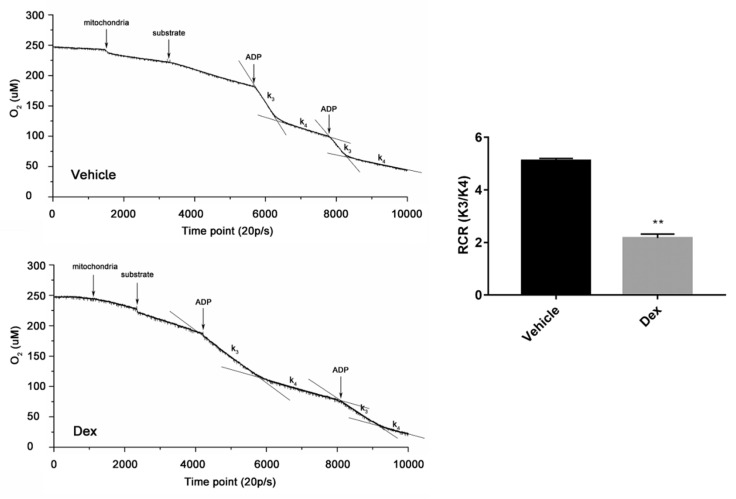
Effect of dexamethasone on respiratory function in mouse liver-isolated mitochondria. After treatment with 1 mM dexamethasone, a Clark oxygen electrode was used to measure oxygen consumption. Data are expressed as Means ± SD, three independent triplicate experiments were performed. ** *p* < 0.01.

**Figure 7 molecules-24-01982-f007:**
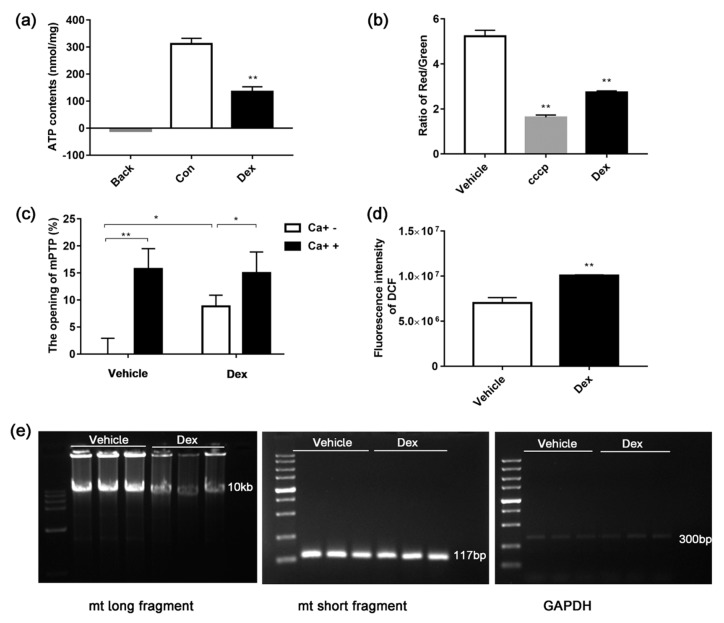
Effect of dexamethasone on mouse liver-isolated mitochondria. The ATP level was determined with a luciferase-based luminescence-enhanced ATP assay kit (**a**). MMP was measured by Multi-Mode Microplate Reader using a fluorescent probe of JC-1 (**b**). The opening of mPTP was measured by detecting the A520 absorbance of mitochondria exposed to 250 µM CaCl_2_, “+”: treatment with Ca^+^; “−“: treatment without Ca^+^ (**c**). ROS was detected by a Multi-Mode Microplate Reader using a fluorescent probe of DCFH-DA (**d**). The mtDNA damage was determined by the ratio of long and short fragments using long PCR (**e**). Data are expressed as Means ± SD, three independent triplicate experiments were performed. * *p* < 0.05, ** *p* < 0.01.

**Table 1 molecules-24-01982-t001:** Sequence of primers for RT-PCR and long PCR.

Target Gene	Primer Sequence	Size (bp)	Accession Numbers
*Mfn1*	Forward: 5′-GCTGTCAGAGCCCATCTTTC-3′ Reverse: 5′-CAGCCCACTGTTTTCCAAAT-3′	195	NM_024200
*Mfn2*	Forward: 5′-GCCAGCTTCCTTGAAGACAC-3′ Reverse: 5′-GCAGAACTTTGTCCCAGAGC-3′	208	NM_001355590
*DRP1*	Forward: 5′-ATGCCTGTGGGCTAATGAAC-3′ Reverse: 5′-AGTTGCCTGTTGTTGGTTCC-3′	180	NM_001360010
*PGC-1α*	Forward: 5′- CGGAAATCATATCCAACCAG-3′ Reverse: 5′-TGAGGACCGCTAGCAAGTTTG-3′	243	XM_006503779
*NRF1*	Forward: 5′- TGGTCCAGAGAGTGCTTGTG-3′ Reverse: 5′- TTCCTGGGAAGGGAGAAGAT-3′	184	NM_001361693
*TFam*	Forward: 5′-GGAATGTGGAGCGTGCTAAAA-3′ Reverse: 5′-TGCTGGAAAAACACTTCGGAATA-3′	118	NM_009360
Long fragment	Forward: 5′-TACTAGTCCGCGAGCCTTCAAAGC-3′ Reverse: 5′-GGGTGATCTTTGTTTGCGGGT-3′	8636	AJ512208.1
Short fragment	Forward: 5′- CCCAGCTACTACCATCATTCAAGT -3′ Reverse: 5′-GATGGTTTGGGAGATTGGTTGATG -3′	117	NC_005089
*β-actin*	Forward: 5′-CCTGAGGCTCTTTTCCAGCC-3′ Reverse: 5′-TAGAGGTCTTTACGGATGTCAACGT-3′	110	NM_007393

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
