# Peer review of "Dexamethasone-Induced Mitochondrial Dysfunction and Insulin Resistance-Study in 3T3-L1 Adipocytes and Mitochondria Isolated from Mouse Liver"

_molecules, 2019, doi:10.3390/molecules24101982_

Round 1

Reviewer 1 Report

Ø  Firstly, I checked the Plagiarism and found it was 56% plagiarism. Please obey with moral ethics and use your own words to re-write your manuscript.

Ø  Line 23-24, “Dexamethasone’ roles in mitochondrial dysfunction which plays an important role in energy metabolism disorders are not well-studied” was not exact. Many published talked about Dexamethasone roles in mitochondrial dysfunction such as “Mol Pharm. 2016 Jan 4;13(1):73-84”. “Cell Physiol Biochem 2018; 49:758–779”.  

Ø  Line 59 to 68, I don’t think this paragraph was necessary because it’s very basic knowledge.

Ø  To understand the aims and innovations of the project, suggest that author descript the relationship among dexamethasone, diabetes and adipose mitochondrial function.

Ø  For consideration of cell line, 3T3-L1 adipocytes, did author checked the mycoplasma contamination? What cell is the vehicle? Adipocytes? Modified adipocytes? If using modified cells, the line 81 to 83 should be changed.

Ø  To be repeated by other scientist, please provide more information regarding procession of 3T3-L1 pre-adipocytes differentiation and insulin resistance induction like briefly bababa….

Ø  Line 91, NBDG should be full name when it was first time to show up.

Ø  I didn’t see the detail regarding NBDG method, please add it.

Ø  Is Akt phosphorylation diagnosed marker for diabetes?

Ø  Line 290, ICR full name?

Ø  Line 294 should provide IACUC number.

Ø  For mitochondrial isolation, I doubted that the buffer containing 250 mM sucrose, 10 mM Tris-HCl and 1 mM EDTA, pH 7.4 was best isolation buffer for getting integrity mitochondria. Please giving more evidences.

Ø  Did the RIPA buffer contain protease/phosphorylase inhibitors?

Ø  Line 344 to line 349, why the author used DCFH-DA to measure mitochondrial ROS production? DCFH-DA was general used to measure intracellular ROS, MitoSOX was applied for mitochondrial ROS.

Ø  Did the author remove contaminated RNA or DNA after isolation DNA or RNA?

Ø  How the author measure cDNA concentration? Because 15 ng cDNA (line 391) was used to run qPCR. How to normalize the PCR production if using mtDNA as template?

Ø  Table 1 should provide the PCR amplified size and genebank access numbers.

Ø  For Long PCR, I 100% doubted the method, one, 8 min couldn’t amply 8636-bp size; two, 40 cycles couldn’t showed the different in regular Long- and/or short PCR except qPCR.

Ø  Line 438, mtNDA should be mtDNA.

Ø  Figure 2A. was difficult to understand except proving higher amplifier imagines and DAPI staining for normalization.

Ø  Suggest that repeat Figure 4b using qPCR and 4c using lower cycles (what cycles are suitable should be determined by a serials PCRs).

Ø  For figure 6, did the author used rotenone to inhibitor complex I-driving ATP production? If not, please repeat it.

Ø  Figure 7d was used DCFH-DA not MitoSOX, why? I doubled Figure 7e even only one integrity mitochondria in Dex-treated sample, the PCR production should be having same dosimetry after 40 cycles. If provide EM date to show all mitochondria got damaged in Dex-treated group, Fig 7e is possible.

Author Response

Dear reviewer:

We would like to express our sincere thanks to you for your comments and suggestions to our manuscript. You have raised many critical questions in your review comments, which will improve the quality of the manuscript and benefit me a lot. We have clearly marked all revisions in the electronic version of the revised manuscript by using a yellow background.

Ø Firstly, I checked the Plagiarism and found it was 56% plagiarism. Please obey with moral ethics and use your own words to re-write your manuscript.

We have kindly checked the repetition rate about the manuscript and found that the materials and methods section was similar to one paper we published earlier (J. Agric. Food Chem. 2018, 66, 3169−3178) due to some of the experimental methods used in these two experiments were the same. We have revised the repeated parts in the revised manuscript and checked by a native English speaking colleague.

Ø Line 23-24, “Dexamethasone’ roles in mitochondrial dysfunction which plays an important role in energy metabolism disorders are not well-studied” was not exact. Many published talked about Dexamethasone’ roles in mitochondrial dysfunction such as “Mol Pharm. 2016 Jan 4;13(1):73-84”. “Cell Physiol Biochem 2018; 49:758–779”.

We have made changes in the revised manuscript.

Ø Line 59 to 68, I don’t think this paragraph was necessary because it’s very basic knowledge.

We have made changes in the revised manuscript.

Ø To understand the aims and innovations of the project, suggest that author descript the relationship among dexamethasone, diabetes and adipose mitochondrial function.

We have made changes in the revised manuscript.

Ø For consideration of cell line, 3T3-L1 adipocytes, did author checked the mycoplasma contamination? What cell is the vehicle? Adipocytes? Modified adipocytes? If using modified cells, the line 81 to 83 should be changed.

The 3T3-L1 pre-adipocytes cell line was purchased from the cell bank of the Institute of Biochemistry and Cell Biology of Shanghai (shanghai, China) and mycoplasma contamination has been detected in the cell bank before shipping. 3T3-L1 adipocytes were induced from 3T3-L1 pre-adipocytes using induced medium I (0.5 mM 3-isobuty-1-methylxanthine, 1 μM dexamethasone, and 10 μg/mL insulin) and induced medium II (10 μg/mL insulin).

Ø To be repeated by other scientist, please provide more information regarding procession of 3T3-L1 pre-adipocytes differentiation and insulin resistance induction like briefly bababa….

We have added the detailed induction method regarding procession of 3T3-L1 pre-adipocytes differentiation and insulin resistance induction in the section 4.1.

Ø Line 91, NBDG should be full name when it was first time to show up.

We have made changes in the revised manuscript.

Ø I didn’t see the detail regarding NBDG method, please add it.

We have added the detailed method about 2- NBDG detection in the section 4.4. 2-NBDG Uptake Assay.

Ø Is Akt phosphorylation diagnosed marker for diabetes?

In this study, we induced 3T3-L1 adipocytes insulin resistance model using 1μM dexamethasone. Insulin resistance is a major risk factor and the most important symptom of type 2 diabetes. When insulin resistance occurs, cells or tissues are less sensitive to insulin and the same dose of insulin does not work. AKT is a key protein in the insulin signaling pathway, and its phosphorylation level is very sensitive to insulin. Under insulin resistance, AKT phosphorylation levels decreased at the same dose of insulin stimulation.

Ø Line 290, ICR full name?

We have made changes in the revised manuscript.

Ø Line 294 should provide IACUC number.

We have provided the animal experiment ethic approval files to the editorial office.

Ø For mitochondrial isolation, I doubted that the buffer containing 250 mM sucrose, 10 mM Tris-HCl and 1 mM EDTA, pH 7.4 was best isolation buffer for getting integrity mitochondria. Please giving more evidences.

The isolation buffer using in this study is based on previous studies (CANCER RESEARCH, 1990, 50, 7876-7881). We lightly adjusted it based on the activity of the isolated mitochondria. And we have also found that the composition of liver mitochondrial isolation buffer sued in many previously literatures (Nature protocols, 2007, 2(2): 287-295; FEBS letters, 1988, 226(2): 265-269.) are similar to those we used.

Ø Did the RIPA buffer contain protease/phosphorylase inhibitors?

The RIPA buffer contains 1 mM PMSF and phosphatase inhibitor 1, 2, 3.

Ø Line 344 to line 349, why the author used DCFH-DA to measure mitochondrial ROS production? DCFH-DA was general used to measure intracellular ROS, MitoSOX was applied for mitochondrial ROS.

We measure intracellular ROS using DCFH-DA and mitochondrial ROS using MitoSOX. MitoSOX is a mitochondria-specific ROS probe and DCFH-DA is capable of detecting total ROS in cells. Line 344 to line 349, we used DCFH-DA to measure isolated mitochondrial ROS production. In this reaction system, there are only isolated mitochondria and no intact cells. Therefore, almost all of the ROS in the system result from mitochondria. So, we used DCFH-DA, a cheaper and more accurate probe, to detect ROS in this system.

Ø Did the author remove contaminated RNA or DNA after isolation DNA or RNA?

Yes, the contaminated RNA or DNA were removed after isolation DNA or RNA. The DNA extraction kit we use contains RNase A and DNase I was used after RNA was extracted using TRIzol.

Ø How the author measure cDNA concentration? Because 15 ng cDNA (line 391) was used to run qPCR. How to normalize the PCR production if using mtDNA as template?

For qPCR, 1μL synthetic cDNA or 15 ng gDNA was used. We have made changes in the revised manuscript. We used mtDNA as template to detect mtDNA copy number and normalized using β-actin.

Ø Table 1 should provide the PCR amplified size and genebank access numbers.

We have added the PCR amplified size and genebank access numbers in Table 1.

Ø For Long PCR, I 100% doubted the method, one, 8 min couldn’t amply 8636-bp size; two, 40 cycles couldn’t showed the different in regular Long- and/or short PCR except qPCR.

For Long PCR, we used LongAmp Taq 2X Master Mix (BioLabs M0287L) to amply 8636-bp mitochondrial fragment and the amplification efficiency of the Taq enzyme is 50s/kb. So, 8 min absolutely could amply 8636-bp size.

Due to our negligence, the cycles of Long- and short PCR were wrong after checking the original record. The Long fragment cycles were 28 and the short fragment cycles were 30. We are very sorry about this and have made changes in the revised manuscript.

Ø Line 438, mtNDA should be mtDNA.

We have made changes in the revised manuscript.

Ø Figure 2A. was difficult to understand except proving higher amplifier imagines and DAPI staining for normalization.

Figure 2A. was replaced with more higher amplifier imagines and the bright field and merged images were added.

Ø Suggest that repeat Figure 4b using qPCR and 4c using lower cycles (what cycles are suitable should be determined by a serials PCRs).

We repeated Figure 4b with qPCR and obtained similar results. The Long fragment cycles were 28 and the short fragment cycles were 30 after checking the original record. We have made changes in the revised manuscript.

Ø For figure 6, did the author used rotenone to inhibitor complex I-driving ATP production? If not, please repeat it.

Yes, the mitochondria were incubated with 2 μM rotenone for 5 min to inhibit complex I activity before succinate was added.

Ø Figure 7d was used DCFH-DA not MitoSOX, why? I doubled Figure 7e even only one integrity mitochondria in Dex-treated sample, the PCR production should be having same dosimetry after 40 cycles. If provide EM date to show all mitochondria got damaged in Dex-treated group, Fig 7e is possible.

In this reaction system, there are only isolated mitochondria and no intact cells. Therefore, almost all of the ROS in the system result from mitochondria. So, we used DCFH-DA, a cheaper and more accurate probe, to detect ROS in this system. The Long fragment cycles were 28 and the short fragment cycles were 30 after checking the original record. We have made changes in the revised manuscript.

Yours Sincerely,

Guangxiang Luan

Reviewer 2 Report

Molecules (molecules-495060), Comments to the Authors:

Title: Mitochondrial Dysfunction Induced in The Process of Dexamethasone Mediated Insulin Resistance in 3T3-L1 Adipocytes and Mouse Liver Isolated Mitochondria

Comments

The submitted manuscript discussed the if mitochondrial dysfunction is involved in the pathogenesis of dexamethasone-induced insulin resistance. An insulin resistant model in 3T3-L1 adipocyte was established by incubating with 1 μM dexamethasone for 48 h, and then the mitochondrial dysfunction was studied. Results showed that dexamethasone impaired insulin-induced glucose uptake increase and mitochondrial function. Dexamethasone treatment caused a decrease in mitochondrial membrane potential (MMP) and intracellular ATP synthesis while induced intracellular and mitochondrial reactive oxygen species (ROS) accumulation. The mtDNA damage was also investigated and the result indicated that dexamethasone treatment led to mtDNA damage. Mitochondrial dynamic changes were observed as decreased expression of Drp1 and increased expression Mfn2 compared with controls. Levels of PGC-1, NRF1 and TFam mRNA expression level were downregulated after dexamethasone treatment. There was no significant change in mitochondrial DNA (mtDNA) copy number while compensatory increases in mitochondrial mass. Dexamethasone effect on isolated mitochondria in mouse liver was also studied. Results shown that in dexamethasone treatment group mitochondrial respiratory function was reduced: mitochondrial respiration controlling rate (RCR) decreased, MMP droped, ATP synthesis ability declined, the opening of mitochondrial permeability transition pore (mPTP) was induced, mtDNA damaged, and ROS accumulated.

The effect of dexamethasone on mitochondrial function was studied in many previous publications. I think the manuscript can be accepted for publication after the authors respond to the following comments:

Did the authors study the effect of dexamethasone on OXPHOS complex activity      and its expression?

Dexamethasone affects cellular pH, did the authors study the effect of      dexamethasone on cellular pH

AMPK/FOXO3 signaling pathway is involved in dexamethasone effect on mitochondrial      dysfunction. Did the authors study the effect of dexamethasone on      AMPK/FOXO3 signaling pathway?

Did the authors study the effect of dexamethasone on cytosolic calcium?

In previous manuscripts, it was shown that dexamethasone affected the following      proteins Cox2, Tom20, UCP3, pDRP1, DRP1 and FIS1 proteins. Did the authors      in the current manuscript investigated the effect of dexamethasone on      these proteins.

Dexamethasone-induced alterations in OPA1, MFN2,      PINK1, and PARKIN levels in previous reports leading to mitochondrial dysfuntction.      Did the authors study the effect of dexamethasone on OPA1, MFN2, PINK1,      and PARKIN levels?

It will improve the quality of the paper if the authors drawn a      schematic diagram showing the molecular targets of dexamethasone in 3T3-L1      adipocytes.

Author Response

Dear reviewer:

We would like to express our sincere thanks to you for your comments and suggestions to our manuscript. You have raised many critical questions in your review comments, which will improve the quality of the manuscript and benefit me a lot. We have clearly marked all revisions in the electronic version of the revised manuscript by using a yellow background.

Ø Did the authors study the effect of dexamethasone on OXPHOS complex activity and its expression?

In previous experimental studies, we used the substrate consumption rate to detect the activity of the OXPHOS complex. However, this indirect method does not directly reflect the activity of the mitochondrial respiratory chain. The efficiency of the mitochondrial respiratory chain requires the cooperation of the all complexes and the integrity of the mitochondrial membranes. In this study, we directly used the mitochondrial respiratory control (RCR) rate to evaluate the efficiency of the mitochondrial respiratory chain.

Ø Dexamethasone affects cellular pH, did the authors study the effect of dexamethasone on cellular pH?

Thank you very much for your valuable comments and suggestions. Cellular pH is a very important factor for cell metabolism. In this study we have not paid attention to the intracellular pH, but in our future study we will focus on the intracellular pH.

Ø AMPK/FOXO3 signaling pathway is involved in dexamethasone effect on mitochondrial dysfunction. Did the authors study the effect of dexamethasone on AMPK/FOXO3 signaling pathway?

We have focused on the AMPK signaling pathway in our previous studies (Life Sciences 136 (2015) 120125) and found that dexamethasone impaired GLUT4 membrane transposition by down-regulating AMPK phosphorylation levels in 3T3-L1 adipocytes.

Ø Did the authors study the effect of dexamethasone on cytosolic calcium?

Thank you very much for your valuable suggestions, we will focus on it in our future study.

Ø In previous manuscripts, it was shown that dexamethasone affected the following proteins Cox2, Tom20, UCP3, pDRP1, DRP1 and FIS1 proteins. Did the authors in the current manuscript investigated the effect of dexamethasone on these proteins?

In order to adapt to energy metabolism in different tissues or different environments, mitochondria have been in a dynamic equilibrium of fusion and division. DRP1 and FIS1 are mitochondrial fission-associated proteins, which play an important role in the dynamic process of fusion and fission of mitochondria. In our research, DRP1 expression level was measured and the result showed that DRP1expression was inhibited by dexamethasone. UCP3 is an important member of the mitochondrial carrier protein family, which is mainly distributed in the mitochondrial inner membrane of skeletal muscle, which mediates the uncoupling of the oxidation process from the ADP phosphorylation process, so that energy is not stored in the form of ATP, but is released in the form of heat. We don't pay much attention to the UCP3 in adipocytes. About Cox2 and Tom20, we are paying attention to our current research.

Ø Dexamethasone-induced alterations in OPA1, MFN2, PINK1, and PARKIN levels in previous reports leading to mitochondrial dysfuntction. Did the authors study the effect of dexamethasone on OPA1, MFN2, PINK1, and PARKIN levels?

OPA1, MFN2 are mitochondrial fusion-associated proteins, which also play an important role in the dynamic process of fusion and fission of mitochondria. MFN1 and MFN2 expression level was increased after dexamethasone treatment. These results about DRP1, MFN1, and MFN2 result that adipocyte mitochondrial dynamic was broken after dexamethasone treatment. PINK1/PARKIN signal pathway will be investigated in our future study.

Ø It will improve the quality of the paper if the authors drawn a schematic diagram showing the molecular targets of dexamethasone in 3T3-L1 adipocytes.

Thank you for your suggestions. Our current research data showed that dexamethasone induced mitochondrial dysfunction and insulin resistance in 3T3-L1 adipocytes and mitochondria isolated from mouse liver. However, these observed results cannot give the direct molecular target of dexamethasone which is studying in our current research. But we have provided a Graphical Abstract to editor office.

Yours Sincerely,

Guangxiang Luan

Round 2

Reviewer 1 Report

the plagiarism is same as prior version, which is big problem and nothing changed. I do think the author should understand that plagiarism is not allowed especially in high qualified journals. What I suggest is to reject it.

Author Response

Dear Reviewer,

In accordance with the suggestion from you, we have carefully revised the manuscript to cut all potential repetition. We especially paid attention to the repletion of sentence and long phrases in all sections including abstract, introduction, results, discussion and methods.

After revision, we also did a repetition check with the “CopySpider” software (under a “deep” mode) and several other free online tools. According to the screening result, we think the repetition is less than 15% (CopySpider gave a result of 10.17% even under “deep” mode and affiliations or references were included). We attached the screening result given by “CopySpider” (please see following pages). Please note that terminology or fixed phase we used should not be considered as repetition.

We sincerely appreciate the help from you to elevate the quality of our manuscript.

Best,                                                                                

Guangxiang and Honglun 

Reviewer 2 Report

Molecules (molecules-495060), Comments to the Authors:

Title: Mitochondrial Dysfunction Induced in The Process of Dexamethasone Mediated Insulin Resistance in 3T3-L1 Adipocytes and Mouse Liver Isolated Mitochondria

Comments:

After reading the revised manuscript and the authors response to my comments, I think the manuscript can be accepted for publication.

Author Response

Dear Reviewer,

We sincerely appreciate the help from you to elevate the quality of our manuscript and your comments of the manuscript can be accepted for publication.

In accordance with the Comments from another reviewer, we have carefully revised the manuscript to cut all potential repetition. We especially paid attention to the repletion of sentence and long phrases in all sections including abstract, introduction, results, discussion and methods.

After revision, we also did a repetition check with the “CopySpider” software (under a “deep” mode) and several other free online tools. According to the screening result, we think the repetition is less than 15% (CopySpider gave a result of 10.17% even under “deep” mode and affiliations or references were included).

Best,                                                                                

Guangxiang and Honglun